# Sniffing speeds up chemical detection by controlling air-flows near sensors

Thomas L. Spencer [1], Adams Clark[1], Jordi Fonollosa [2,3,4], Emmanuel Virot [5] & David L. Hu [1,6✉]

Most mammals sniff to detect odors, but little is known how the periodic inhale and exhale that make up a sniff helps to improve odor detection. In this combined experimental and theoretical study, we use fluid mechanics and machine olfaction to rationalize the benefits of sniffing at different rates. We design and build a bellows and sensor system to detect the change in current as a function of odor concentration. A fast sniff enables quick odor recognition, but too fast a sniff makes the amplitude of the signal comparable to noise. A slow sniff increases signal amplitude but delays its transmission. This trade-off may inspire the design of future devices that can actively modulate their sniffing frequency according to different odors.

[1] School of Mechanical Engineering, Georgia Institute of Technology, Atlanta GA, USA. [2] B2SLab, Departament d'Enginyeria de Sistemes, Automàtica i Informàtica Industrial, Universitat Politècnica de Catalunya, 08028 Barcelona, Spain. [3] Networking Biomedical Research Centre in Bioengineering, Biomaterials and Nanomedicine (CIBER-BBN), Madrid, Spain. [4] Institut de Recerca Sant Joan de Déu, 08950 Esplugues de Llobregat, Spain. [5] John A, Paulson School of Engineering and Applied Sciences, Harvard University, Cambridge, Massachusetts MA, USA. [6] School of Biology, Georgia Institute of Technology, Atlanta GA, USA. ✉email: hu@me.gatech.edu

Dogs are well known for their excellent sense of smell, and working dogs are still the primary way of detecting odors in uncertain environments such as drug detection in airports and cancer detection in hospitals. However, using dogs as chemical detectors is expensive and at times unreliable[1]. Consequently, much work has been done on designing and building electronic noses[2]. Improving electronic noses involves improving either the sensor or the system that delivers odors to the sensor. Commonly, a sensor is heated in order to broaden the response of the system[3,4]. Odor delivery is a challenging problem and several investigators have designed biologically-inspired devices to deliver the odors. Staymates et al. used a 3D printed dog's nose to inhale and exhale odors, which improved the detection due to drawing in airflows directly from the source[5]. Kohnotoh et al. used dual air pumps to imitate an inhale from the nostrils, determining directionality from the difference in response[6]. In this study, we build a sniffing device to visualize and measure the benefits of sniffing at different frequencies.

Using a device to study sniffing can give insight into biology. Although most mammals sniff, a clear benefit for this behavior is still missing. Potential hypotheses include creating turbulence to better mix the odor and providing repeated trials for identifying and confirming odors[7]. By far, the majority of olfaction studies have been on animals such as rodents and dogs. Studies of dogs of different sizes suggest that sniffing frequency ranges from 4–8 Hz and does not change systematically with body size[5,8–10]. However, other animals such as mice[11–14] sniff at up to 12 Hz. On the other size extreme, elephants are extremely adept with their olfactory system and are being used to detect substances at low concentration such as TNT[15], yet their sniff frequency had never been measured. In this study, we present a scaling law for sniffing that incorporates a larger range of body mass. We also present a series of mathematical models based on the pressure, compliance, and turbulence within the respiratory system to rationalize the scaling of sniff frequency with body size.

The data from our mechanical sniffer is rationalized through a fluid mechanics model of odor detection. Our model builds upon previous theoretical solutions for oscillatory flow in an artery, derived by British mathematician John R. Womersley at the advent of cardiovascular fluid mechanics studies in the 1950's[16]. Previous models of sniffing flows have generally been computational[17] rather than analytical[18], and have relied on the complex nasal cavity system in the animal. However, few models can incorporate the complexity of biological nasal passages, the chemical-sensor response, and the fluid mechanics of the air flow, each difficult problems in their own right. Building biologically-inspired devices like ours can be an important first step in testing biological hypotheses, verifying theoretical studies, and detecting odors quickly and reliably.

In this work, we show sniffing airflows can improve the speed and amplitude of the signals measured. We pay particular attention to a dimensionless group called the Womersley number that takes into account the width of the channel and the frequency of sniffing. High-frequency sniffing is useful for both sensors and animals because it obtains data faster than a single inhale of air. Our bellows-driven system GROMIT, along with our theoretical model, demonstrates that choosing a frequency of sniffing should consider trade-offs. The faster the sniff, the lower the signal per sniff but the more data per unit time. Choosing sniff frequency should thus depend on whether the user is trying to maximize data or obtain data as quickly as possible.

## Results

We measure the sniffing dynamics of mammals, from mice to elephants, with details given in the experimental methods section. We use a specially prepared food box and microphone setup to solicit sniffs from an African elephant at Zoo Atlanta, measuring three sniffs from 21 attempts. We combine these rates with data from previous literature, including one study on dogs[9], three studies on rodents[9,13,19], one study on rabbits[20], and one study on shrews[21]. YouTube videos provided two more data points, including a horse and giraffe, whose masses are assumed to be those of adult animals. Figure 1 shows examples of the audio waveforms of sniffing for a rat, dog, elephant, and giraffe respectively. The animals sniffing frequency decreases with increasing body size, from 8 Hz for a rat, 5 Hz for a dog, to 2 Hz for the giraffe and elephant.

Figure 2a shows the relation between sniffing frequency and body mass. The solid black line indicates the power law best fit for the experimental data,

$$f = 8M^{-0.18}, \qquad (1)$$

where maximum observed frequency $f$ is in Hz and body mass $M$ is in kg ($N = 16$, $R^2 = 0.85$). In the math methods section, we compare this experimental trend to predictions from four theoretical models, designated $f_i$, and marked by the blue lines in the figure.

In models $f_1$ and $f_2$ we consider inertial effects. We predict $f_1$ through consideration of geometry of the nasal passages. In $f_2$ we use Leith's 1960's experimental measurements[22] of the compliance and inertance of a range of animal respiratory systems to determine their resonant frequency. Although $f_1$ and $f_2$ use independent data sets, they result in predictions that both compare favorably with the experimental trend, indicating the importance of considering inertial effects.

We consider the effects of viscosity in models $f_3$ and $f_4$. In model $f_3$, we provide a limiting sniffing frequency related to the

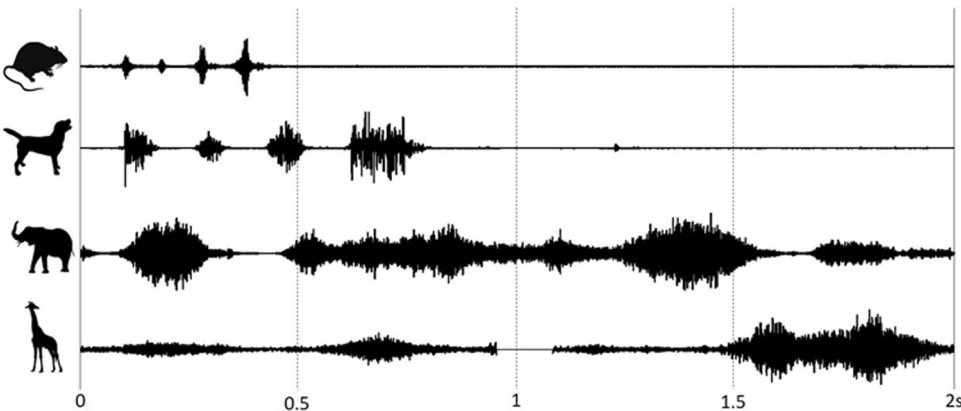

**Fig. 1 Audio waveforms of sniff cycles for a rat, dog, elephant, and giraffe respectively.** Animal silhouettes are from freepik.com.

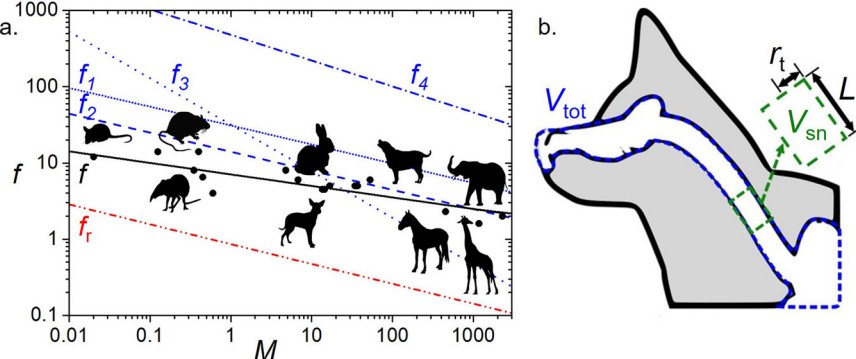

**Fig. 2 Biological sniffing. a** The relationship between maximum observed sniff frequency $f$ (Hz) and body mass $M$ (kg). In addition to elephant sniffing trials conducted here, black data points are from dogs, rodents, rabbits, and shrews from previous studies as well as a rat, dog, horse, and giraffe from YouTube. The black solid line is the power law best fit to the experiments. For comparison we include four theoretical predictions $f_i$ using blue lines and the respiratory rate shown by the red dot-dot-dash line. Animal silhouettes are from freepik.com. **b** Profile of a dog's nose, with the white cavities denoting the nose, nasal cavity, trachea, and lungs, moving from left to right. Here, $V_{tot}$ (in short dash blue) is the total volume of air which must be accelerated each sniff, $V_{sn}$ (in long dash green) is the sniffing tidal volume of new air brought into the respiratory system, $r_t$ is the trachea hydraulic radius, and $L$ is the change in position of the sniffing front.

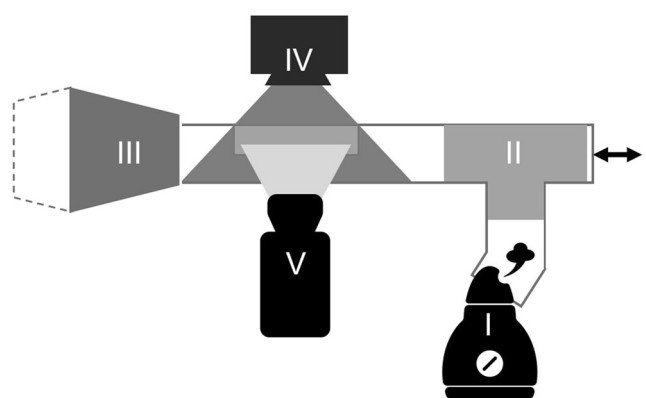

**Fig. 3 Visualization experiment, where humidifier (I) pushes humid air into a rectangular channel through a tee junction (II).** Flow is driven in an oscillatory motion by a custom diaphragm pump (III), illuminated by laser light sheet (IV), and filmed by a high speed camera (V).

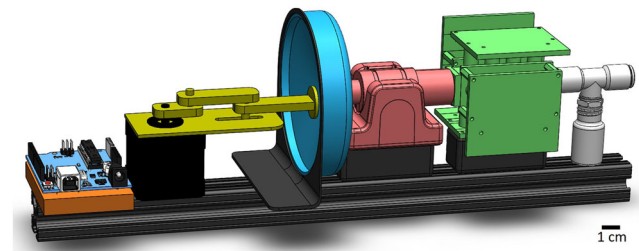

**Fig. 4 GROMIT device schematic with motor controller in orange, motor in black, rotational to linear motion converter in yellow, custom 3D printed diaphragm in blue, flow sensor in red, metal oxide sensor array with conditioning electronics in green, and tee junction with sample bottle in white.** Scale bar represents 1 cm.

maximum Reynolds number[23] associated with laminar flows. Animals below 30 kg fall below the trend line $f_3$, indicating animals of this size or lower experience laminar flows during sniffing. Model $f_4$ is derived from relations between flow rate and Womersely number. The poor fit of model $f_4$ may be due to the use of an infinite tube to characterize the finite-length trachea.

For further comparison, we also include a red line indicating the frequency of relaxed breathing $f_r = 0.89M^{-0.26}$ ($N = 692$) reported by Stahl in 1967[24]. Comparing the black line to the red line indicates that sniffing occurs 10 times faster than breathing. The exponents for the sniffing, breathing, and the theory (models $f_1$ and $f_2$) have comparable values (from $-0.18$ to $-0.26$), suggesting that the dominant physics we have proposed is correct and has predictive power. To understand why animals sniff faster than they breathe, we turn to flow visualization of sniffing.

We design and build a bellows system that we call GROMIT, or Gaseous Recognition Oscillatory Machine Integrating Technology, which imitates a sniff by sampling ethanol vapor at set frequencies. More detail on GROMIT is given in the experimental methods. In Fig. 3, we show a schematic where GROMIT is combined with humid air and a laser light sheet to visualize the airflows resulting from sniffing. Figure 4 shows a schematic representing the different aspects of the system with its integrated sensors.

The airflow of a sniff is characterized by the hydraulic radius of the ethmoidal chamber $D_h/2$, the frequency of oscillation $f$, and the kinematic viscosity of the air $\nu$. Together, the dimensionless group that characterizes the system is called the Womersley number, Wo, which was first used to describe cardiovascular flows[25,26] and may be written

$$\text{Wo} = D_h\sqrt{\frac{\pi f}{2\nu}}. \tag{2}$$

We define Wormesley number in terms of half the hydraulic diameter to stay consistent with the theory derived by Womersley[16]. Higher Womersley numbers are associated with higher relative magnitudes of inertia compared to viscous forces. We identified four animals for which the Womersley number has been measured throughout the nasal cavity: these include a pygmy marmoset[27], rabbit[20], dog[8], and deer[28], which have Womersley numbers of 0.2–2.5. The pygmy marmoset has a mass of 0.1 kg whereas the deer weighs nearly three orders of magnitude more at 60 kg, indicating that Womersley number changes slowly with body mass. We perform our device testing within the range of Wo from 0.5 to 7.5 to try to encompass the observed biological range. Reporting Womersley for animals requires CT-scanning to map the complex nasal cavity. These maps are reduced to a simplified shape using the hydraulic diameter, which takes into account both the cross sectional area and the perimeter

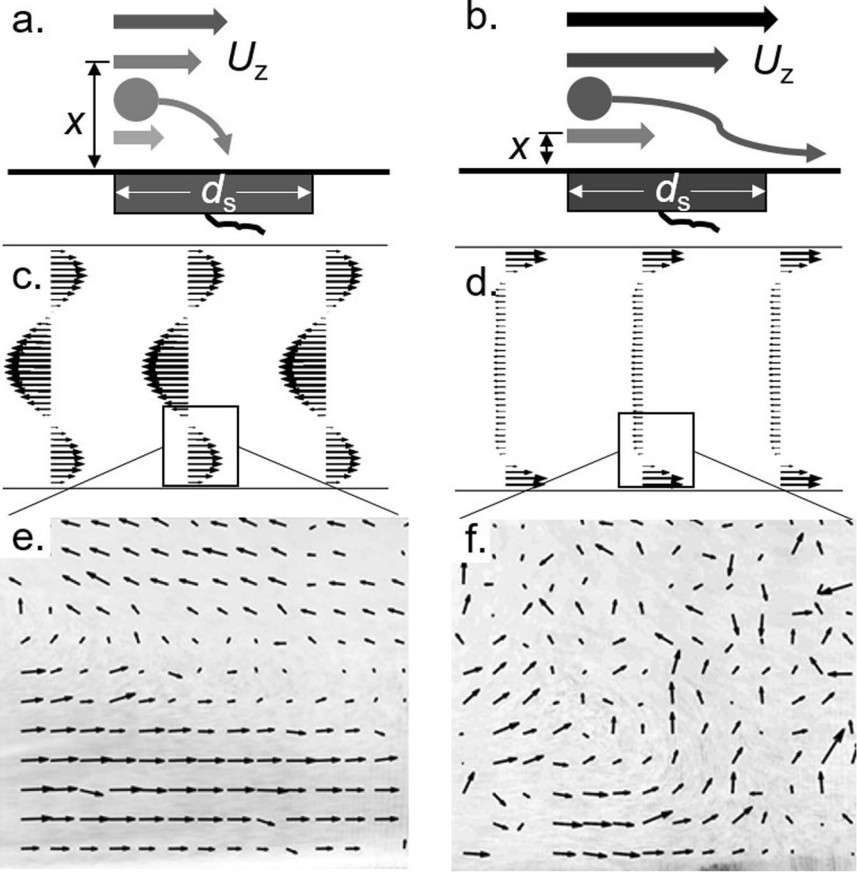

**Fig. 5 Flows created by sniffing. a–b** Illustrations showing a particle in low Womersley flow (**a**) has a better chance to strike the sensor surface than in high Womersley flow (**b**). $U_z$ is the axial velocity, $x$ is the diffusion distance, and $d_s$ is the sensor diameter. **c–d** Simulated flow profile at transition from inhalation to exhalation for Wo = 1.5 and Wo = 7.5, respectively. **e–f** Particle image velocimetry of flow profile at transition from inhalation to exhalation for low and high Wo respectively.

of the chamber[29]. Craven et al.[8] define hydraulic diameter as

$$D_h = \frac{4A_c}{P} \qquad (3)$$

where $A_c$ is the cumulative surface area of the olfactory surface calculated with finite difference numerical integration and $P$ is the perimeter of the cross section of interest. Thus, a complicated olfactory structure can be reduced to a cylindrical shape for ease of calculation of the associated flows.

In GROMIT, flows are generally unidirectional with respect to the tube axis, with a velocity that varies with time and distance from the tube wall. We illustrate these properties with the flow field at the transition from inhalation to exhalation, where the applied pressure transitions from positive to negative. Figure 5e, f show the observed velocity field for Womersley numbers of 1.5 and 7.5. As indicated by the arrows, the flow at the walls changes direction before the midstream flow does. A model for molecular deposition on a sensor should thus take the time and space varying flows into account. These flows are comparable to those generated with COMSOL Multiphysics to solve for the flow due to an oscillatory pressure, as shown in Fig. 5c, d.

We present a mathematical model in the Supplementary Information that predicts the flow velocity in a circular channel, which we use to approximate our experiments in a square channel. A key feature of our model is tracking the diffusion time of molecules given by the ratio of the sensor width $d_s = 5$ mm and the particle's speed, which is directed parallel to the face of the sensor. The slower the air speed, the more time the molecules

spend in the vicinity of the sensor, and in turn, the closer to the sensor they may travel by diffusion. In our model, Womersley theory gives a closed form solution for the velocity field as a function of distance from the sensor. We discretize the sniffing cycle into discrete time points. At each time point, we calculate a diffusion time and the maximum distance from the sensor that particles can still diffuse onto the sensor. We then integrate across all points in the cycle to determine the net diffusion of molecules onto the sensor.

In short, our model is a quasi-steady diffusive model where odor is advected according to Womersley flow, but then accrues on the sensor by diffusion. There is no time-dependence on the concentration field. Because the diffusion time scale is much longer than the convective time scale, only a thin layer of the air volume has a chance to diffuse onto the sensor. For our channel of 1 cm radius, the layer of detected air is only 0.35 mm thick.

To characterize the sensor response to different Womersley numbers, we conduct tests with ethanol. Figure 6a shows the time course of sensor current (mA) for an ethanol concentration of 89 parts per thousand. Ethanol reacts with the sensing layer by causing oxide ions to release electrons, thereby reducing the sensor's resistance and increasing the flow of current.

The trial begins with GROMIT sniffing using motion of its bellows. The sniffing motions are performed for 30 sec without exposure to ethanol. In this time frame, clearly there is no change in current because the air is empty of ethanol. When ethanol is introduced, the sniffing continues, and the current increases from 10 to 20 mA. Current oscillations of amplitude $A$ are

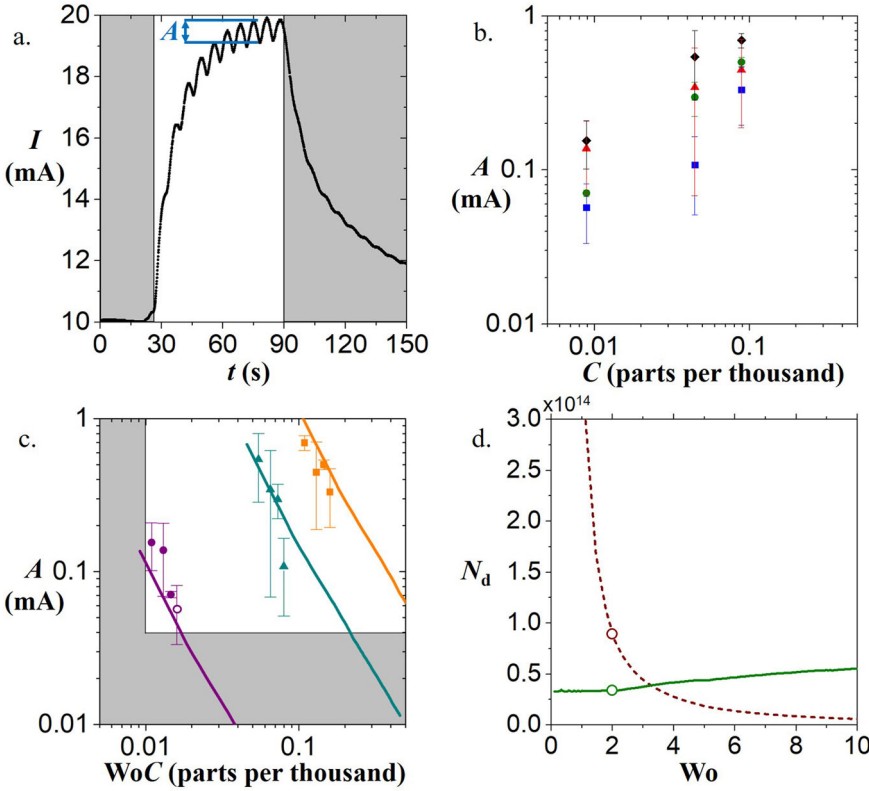

**Fig. 6 Sensor response due to sniffing airflows. a** Time course $t$ of the current $I$ across sensor due to introduction of 89 parts per thousand concentration of ethanol at $t = 30$ sec. In the shaded regions, the ethanol sample bottle is absent from the space; in the remaining regions, the ethanol is present. **b** Relationship between estimated concentration $C$ of ethanol in headspace and current amplitude $A$. Black diamonds, red triangles, green circles, and blue squares indicate response for frequencies of 0.14, 0.2, 0.25, and 0.3 Hz with associated Womersley numbers of 1.25, 1.5, 1.65, and 1.8 respectively. Error bars represent one standard deviation (s.d.). **c** Relationship between amplitude $A$ and product of Womersley number Wo and concentration $C$. Data collected at concentrations of 8.9 parts per thousand, 44 parts per thousand, and 89 parts per thousand represented by the purple circles, cyan triangles, and orange squares respectively. Mathematical models for concentrations of 8.9 parts per thousand, 44 parts per thousand, and 89 parts per thousand represented by the purple, cyan, and orange solid lines respectively. Shaded region shows where no additional information can be obtained by sniffing. At a WoC of less than 0.01, the response returns to baseline each sniff. Below 0.03 mA, no measurable response can be obtained due to the signal dropping underneath the noise threshold of the sensor. Open purple circle represents the optimal collection rates for the 8.9 parts per thousand concentration tests with the chosen sensors. Error bars represent one standard deviation (s.d.). **d** The relationship between number of available molecules for collection $N_d$ and Womersley number. Dashed red line indicates number of molecules available per cycle. Solid green line indicates the number of molecules available per second. Open red and green circles represents collection rates for 8.9 parts per thousand concentration.

synchronized with the motion of the bellows. At a time of 90 sec, ethanol is removed and the current gradually decreases to a baseline value as the ethanol is evacuated from the sensor. Of these features, the initial increase in current due to odor exposure is a standard feature used by many[30-33] to identify an odor. However, without sniffing, this feature requires a time scale on the order of a minute to obtain useful information, which is too long to be useful for animals on the move. In comparison, sniffing brings information to the animal on a $2\pi/f$ time scale, which for our sensor is 7 sec. The increased speed of information transfer may be one reason that high-speed sniffing evolved in animals.

We perform 48 experiments, consisting of 16 tests for each of three different ethanol concentrations (8.9, 44, and 89 parts per thousand). Figure 6b shows the relationship between the amplitude of the sensor current and the concentration of ethanol. The closed symbols (black diamonds, red triangles, green circles, and blue squares) represent the current amplitude for different sniffing frequencies (0.14, 0.2, 0.25, and 0.3 Hz, respectively). The resulting amplitude increases with increasing concentration of ethanol and decreasing frequency, in accordance with the theory presented in the Supplementary Information mathematical modeling section.

We briefly discuss a caveat with regards to the generality of our theoretical model. Our theory assumes the target odor is diffusive in the chosen flow medium such as air. Diffusion enables the odor to leave the streamline and land on the sensor. The ability for an odor to diffuse is characterized by the dimensionless Schmidt number Sc given by the equation

$$\text{Sc} = \frac{\nu}{D} \qquad (4)$$

where $\nu$ is the kinematic viscosity of the fluid and $D$ is the mass diffusivity coefficient. We used ethanol because it is a commonly used chemical and is easily available for testing oxide sensors. For ethanol vapor in air, with a mass diffusivity $D = 11 \times 10^{-6}\,\text{m}^2\,\text{s}^{-1}$ and kinematic viscosity $\nu = 1.48 \times 10^{-5}\,\text{m}^2\,\text{s}^{-1}$, the Schmidt number is 1.4 and within the diffusive regime[34]. Future workers who wish to apply our model will need to use target chemicals for which[34] Sc < 4. According to the Schmidt number, large particles such as dust do not diffuse sufficiently and would require a different technique to capture than the one featured here.

Diffusion can also lead to other effects such as Taylor-Aris dispersion which tends to stretch the distribution of molecules as it travels down the axis of the tube[35]. According to this physical

picture, the current amplitude would decrease and the period would increase over multiple sniffs. However, over a series of ten sniffs, we find that the period and the amplitude remain constant. Thus, we conclude that at the speeds and geometries of our system, Taylor-Aris dispersion is negligible. Future apparatuses using different speeds or chemicals may encounter this effect and are advised to turn to the work of Smith[36] and Ng[37] for interpreting their results.

We now apply our theory to predict the current amplitude for all 48 experiments. We do so by considering the new variable, WoC, the product of Womersley number and ethanol concentration. Figure 6c shows the relationship between sensor current and WoC across three different ethanol concentrations. The solid lines are theoretical predictions given by equation 16 in the Supplement. This prediction relies on a single fitting factor $\beta$ which relates the sensor current to the number of ethanol molecules in the air received. The resulting graph is inherently dimensional due to the sensor response's dependence on input concentration. The theory fits the experiments quite well. The basic trend that can be seen is that signal amplitude $A$ decreases with increasing Womersley number. We may use this trend to understand how both animals and devices might optimize sniffing to maximize the amount of information or the rate of the information transfer.

**Optimal sniffing**. Increasing the frequency of sniffing decreases the duration of the first sniff, enabling data to be obtained more quickly. However, there is a tradeoff. The higher the frequency, the higher the Womersley number, the lower the amplitude of the response (Fig. 6c). If the frequency is increased further, ultimately the amplitude of the response will become so low that it is indistinguishable from noise. For the metal oxide sensor in our study, this noise limit is ~0.03 mA as indicated by the horizontal gray region at the bottom of Fig. 6c.

Too slow a sniff has other downsides as well. A slow sniff can be considered as a continuous flow trial, whose effectiveness has been studied in the past[38]. Because the signal returns to a baseline value in each sniff cycle, no new information is obtained beyond a simple inhale. For example, if the sniff in Fig. 6a was too slow, the current would simply increase from 10 to 20 mA for each sniff, and the amplitude $A$ would equal the traditionally measured total magnitude change, here, 10 mA. The slow sniff limit occurs at a value of 0.01 parts per thousand for the product of Wo (Womersley number) and C (concentration) as indicated by the left-hand shaded region of Fig. 6c.

A similar optimization problem arises when considering the information per sniff (time scale $T = 1/f$) and the information per unit time (time scale $T = 1$ sec). We address this problem using our theoretical model. The relationship between the available molecules $N_d$ and the Womersley number is shown in Fig. 6d. The dashed red curve is the number of molecules per sniff which exponentially decreases and the solid green curve is the number of molecules per second which is roughly linear. If an animal wants to maximize information per unit time, it should sniff at higher Wo. This indeed might be what animals do: when exposed to new odors,[13] mice and rats increase their sniffing frequency by up to 75%. This behavior increases Womersley numbers from 2 to 2.8 and according to our theory, and increases the information per unit time by 10%.

For the specific sensors chosen in this study, and an 8.9 parts per thousand ethanol environment, the highest frequency that can be sniffed is 0.3 Hz, which corresponds to the open purple circle data point in Fig. 6c at Wo = 1.8. At this rate, the device is sniffing as fast as possible to maximize the total number of molecules per second but still generates a signal above noise based

on parameters for our experiments. The expected number of available molecules for collection is $1.6 \times 10^{14}$ molecules per sniff as shown by the open red circle in Fig. 6d and $0.34 \times 10^{14}$ molecules per second as shown by the open green circle in Fig. 6d. These circles indicated the desired regime for sniffing with the chosen sensor. Future designers of electronic nose systems could use sniffing in the appropriate regime to improve their devices for applications such as detecting fruit ripeness[39] and other odor-based tasks[40]. Such designs should adhere to the same trade-off for sniffing: sniff fast enough to get the most information per unit time, but not so fast that the signal per cycle disappears into noise.

Our derivation of an optimal sniffing frequency is valid for a single target chemical. In nature, there are often several target chemicals, each with their own optimal frequency. This is another reason why animals may need to modulate their sniffing frequency in response to unknown odors.

## Discussion

In our study, we used sensor measurements to show that sniffing can improve the acquisition of olfactory data. We hope our work inspires improvements in sensor design, and gives insight into biological sniffing. The results from our study on a simplified model system however should be applied with care to the more complex systems of animals. Animals' noses consist of a complex system of turbinates, whereas in comparison, our system simply consists of a single tube. Our system thus neglects the complex geometry, the benefits of which are yet to be understood.

Our tests only use ethanol, a single source of chemicals, whereas in nature, odors will be combinations of different chemicals. Additionally, our device does not have a liquid coating analogous to the mucus of the biological nose which would require even further time for odorant molecules to diffuse through. In natural noses, different odors land on different portions of the olfactory epithelium, an effect termed odorant partitioning or differential sorption[41]. Our study assumes a uniform of collection of odors along the channel. This assumption is most relevant for relatively insoluble odors, which are deposited rather uniformly along mucus-lined olfactory airways[10].

Our theory showed that increasing sniffing frequency brings more molecules to the sensors' vicinity per unit time. In fact, mice and rats increase their sniffing frequency when exposed to new odors[12,13]. Thus, neurological decoding mechanisms may rely on bringing more total molecules per unit time rather than per unit sniff. For insects, odor stimuli can be resolved on the order of 100 Hz[42]. If the neurological response of the mammalian system is as fast, then sensor sensitivity may not be the bottleneck for maximizing sniff rate.

There may be other constraints on sniffing which we discuss here. For a sniff to be recognized, the sniff must be sufficiently long that odorants can travel to the rear of the nasal cavity where they will be sensed. This constraint may place a constraint on sniffing frequency. For example, dogs have an average velocity through the nasal cavity on the order of $U = 5$ ms$^{-1}$[19,10], and a snout length on the order of $L \approx 0.1$ m[43]. For the sniff to reach the sensory region, the inhalation duration must be at least $L/U = 0.02$ sec, and the period must be $2L/U = 0.04$ sec. This ensures the frequency must be less than 25 Hz which is 3–5 times faster than observed frequency of dogs, 4–8 Hz. We thus conclude that sniffing airflows have plenty of time to reach the position that they are sensed.

Maximum sniffing frequency in animals may also be limited by the ability of odors to diffuse through the mucus layer. Mucus in the nose protects the sensors and serves as a self-cleaning barrier to outside particles, viruses and bacteria. The mucus of a dog's nose is ~10 μm thick[44–47] and has negligible influence on the fluid

motion of the air[10]. However, the diffusion coefficient of molecules through mucus ranges from $6.5 \times 10^{-10}$ to $8.2 \times 10^{-10}\,\mathrm{m^2\,s^{-1}}$, whereas diffusion in the air is of the order $10^{-6}\,\mathrm{m^2\,s^{-1}}$. Given a diffusion distance equal to the dog's mucus thickness, $x = 10\,\mu\mathrm{m}$, Supplemental equation 4 can be solved for the travel time $t_\mathrm{s} = 0.06 - 0.08$ sec, which is associated with frequencies of 12–17 Hz. This mucus diffusion frequency is faster than the sniffing frequency of dogs which ranges from 4–8 Hz. Thus, the bottleneck for information gathering during the sniff is not in the mucus but rather in the flow of information through the air. This idea is consistent with experiments by Uchida and Mainen[48], which show that rats can recognize odors as quickly as the first sniff.

Energetic constraints may give another limit to sniffing frequency. Crawford et al. found the dog's respiratory system has a natural resonant frequency of approximately the same frequency as panting and sniffing at 5 Hz[49]. Seven years later, Spells[50] proposed a respiratory system scaling which can be taken as a (damped) spring-mass oscillator to scale as $f \sim M^{-0.19}$. If the sniff frequency deviates too high above this natural frequency then it could be too energetically costly to maintain[9]. The energetic exertion may be one reason why quicker bouts of sniffing are only observed for a limited time[12,13]. We did not include these energetic constraints in our model since the electronic nose mimic is made with a diaphragm of plastic and latex exhibiting a different resonant frequency to that of a natural respiratory system.

Additionally, at higher frequencies above $\mathrm{Wo} = 1$, the same applied pulsatile pressure difference induces a reduced flow rate (and thus volume of air inspired)[18]. However, in the experiments with GROMIT, we maintain the amplitude of the physical plunger through direct control of the stepper motor. This thereby ensures that the same volume of air is inspired and expired each cycle, independent of sniff frequency. Therefore, in the mathematical model, we ignore the losses in the volume flow rate. However, since the stepper motor must pull harder as Wo increases, the power required to do so may ultimately limit the highest frequencies of future devices.

We utilized low-cost chemical sensors for this study. While beneficial for wide adoption of our techniques, the sensitivity of the sensors required relatively high ethanol concentrations. Additionally, our measurement of current amplitude could be influenced by a number of factors, such as humidity[51,52], pressure[53], temperature[3,4], and mean flow rate[54,55]. Therefore, the order of measurements were reversed on alternating days to ensure the trends were independent of humidity, pressure, and temperature. Lastly, in our mathematical derivation, we assume a circular cross section for the channel. In our experiments, however, the channel is a square cross section because the sensors are flat. Also, our theory assumes fully developed flow, yet our experiments use a channel that is 4 diameters long, which is less than the requirement of 10 diameters to ensure a fully developed profile[56]. Future workers may weigh the costs and benefits of increasing their channel length.

One way to increase the signal to noise ratio of the sensors would be to simply average the response over multiple sniffs[57]. If the noise is additive, random, and zero-mean, then averaging over multiple sniffs should allow for positive noise to cancel with negative noise. This could theoretically lower the lower cut-off indicated by the lower shaded region of Fig. 6c. To apply such a solution, the noise needs to be random and centered in the signal mean. We only measure positive current, so the noise is not the same below the signal than above the signal. Therefore, in this case, averaging the signal will not work, in particular when the signal amplitude is similar (and lower) than the noise level. Additionally, for dynamically changing environments such as are experienced by animals, repeated measurements are often not possible. However, it is possible that animals might use

such methods to get above the noise thresholds of their own sensors.

## Methods

**Elephant sniffing and YouTube sound analysis.** Sound recordings of a 35-year-old female African Elephant (*Loxodonia africana*) of mass 3360 kg and height 2.6 m were taken at Zoo Atlanta in the fall of 2018. We conducted experiments indoors at the edge of the elephant's enclosure in the mornings before the zoo opened to the public. All experiments were guided by the staff at Zoo Atlanta without any direct contact by the authors.

A subdivided 30 cm × 60 cm padded box was placed at a location ~1 m outside the enclosure where the elephant could not visually see the box due to obstruction by the bars of the enclosure. For each trial, bran cubes were placed at a different location every time, in position inside or outside the box. The curators instructed the elephant to reach for and find the food. The elephant employed multiple strategies to find the food including sweeping its trunk side to side in the box as well as sniffing for the food. In the trials where the elephant used predominately sniffing to search, the inhalations and exhalations were recorded with a Blue Yeti-series Snowball microphone, similar to methods used with dogs[58]. The most distinct audio waveform was produced in a trial where the food was placed behind a circular cutout just smaller than the elephant trunk tip diameter. Out of 20 trials, three sound recordings were clear enough for the sniffing bouts to be distinguished from the sound of the trunk knocking into the walls of the box.

The sound recordings of each sniffing bout, including the elephant experiments and non-elephant third party YouTube videos, were manipulated using Audacity's noise reduction effect to reduce the background noise by ~20 dB. The maximum number of peaks in the amplitude per second corresponding to an audible sniff was used to calculate the sniffing frequency. Videos of a horse, giraffe, rat, and dog were analyzed using this method. The maximum sniffing frequency of the rat and dog were confirmed with points from the literature[9,13,13,19].

All experiments were performed in accordance with relevant guidelines and regulations. The Georgia Tech Institutional Animal Care and Use Committee approved protocol number A18068 entitled, "Elephant Sniffing, Breathing, and Suction" for dates November 12, 2018–November 12, 2019.

**Gaseous Recognition Oscillatory Machine Integrating Technology (GROMIT).** We designed and fabricated a sniffing device named the Gaseous Recognition Oscillatory Machine Integrating Technology (GROMIT) which mimics the sniffing mechanics of mammals. The device is designed in modular sections for maximum adaptability. The sections include a custom 3D printed PLA plastic diaphragm pump with a rubber membrane, a Sensirion Venturi flow meter, a custom 3D printed PLA plastic sample housing, and four printed circuit boards with 4 Figaro TGS 2610 sensors on each board.

A schematic of the device can be seen in Fig. 4. A sniff begins with commands from an Arduino Uno microcontroller to a motor controller which in turn sends commands to an Anaheim Automation 15Y2025-LW4 stepper motor. The motor's axial motion is converted from rotational to linear actuation using a custom slider-crank mechanism which is the driving force behind a 3D printed diaphragm pump. The diaphragm pump is shaped in a way to generate the same amount of volumetric flow rate per actuation, thereby mimicking the ability of a mammal's lungs to expand and contract.

By conservation of mass, the relationship between the desired air velocity and system geometry is

$$\delta V_\mathrm{b} = \frac{U_\mathrm{max} A_\mathrm{t}}{f}, \qquad (5)$$

where $A_\mathrm{t}$ is the cross-sectional area of the tubing, $\delta V_\mathrm{b}$ is the volume change of the bellows, $U_\mathrm{max}$ is the desired maximum air velocity, and $f$ is the desired frequency of sniffing between 0.1 and 10 Hz. The bellows volume and tubing area were designed so that the Womersley number of the flow could be modified between 0.5 and 7.5 to represent almost the full range found in mammals.

On the other side of the pump is a Sensirion Venturi flow meter which tracks and verifies the input flow oscillations. The flow meter confirms a one-to-one correspondence between the flow velocity and the input motor signal. The flow sensor is also used to ensure the same average flow rate is obtained for each trial. Next is a series of Figaro metal oxide sensors that were powered on at least 1 week before its first use in order to heat up and remove contaminants, a process called burn in. The sensor section incorporates 4 TGS sensors per board in series for a total of 16 sensors. The board draws ~700–800 mA and is kept powered on before and after each measurement test to elude transient unstable response that appears when power is applied to the sensor when it was unpowered for some time[59,60]. In our experiments, we run the sensors at a voltage of 5.6 V. Last is a section for the test sample to be placed where the headspace is in series with the flow. In order to avoid unwanted dead spaces in the flow path, the tubing cross sectional area was designed to be constant across all sections. The dead volume in the tubing is estimated to be on the order of 60 mL.

We mixed our odor source before conducting trials. We performed experiments with three concentrations: 1 part ethanol to 10 parts DI water by volume, equal parts ethanol and DI water, and pure ethanol solutions. To determine the concentration,

Henry's law $C = \frac{\rho_e}{uHP_{atm}}$ was utilized with ethanol density $\rho_e = 789$ kg m$^{-3}$, ethanol atomic mass $u = 0.04607$ kg mol$^{-1}$, Henry's constant $H = 1.9$ mol m$^{-3}$ Pa$^{-1}$, and pressure $P_{atm} = 1$ atm $= 1,01,325$ Pa[34]. Using this conversion, the concentration levels tested were 8.9 to 89 parts per thousand. These reported concentrations are estimated at the inlet and provide upper limits for the expected lower concentrations which make it to the sensors themselves.

**Flow Visualization**. Tracer particles were generated in the form of humid air generated by a Crane humidifier model number EE-5301. The humid air was introduced into the entrance of the flow using a tee junction to ensure no net momentum was added to the oscillating flow pattern of the sniffing device, Fig. 3. A rectangular channel was built with approximately the same cross sectional dimensions as the rest of the tubing in order to maintain unidirectional flow. A section of the channel was removed and replaced with an optically clear acrylic section with toothpaste applied to the inside to prevent fogging. A Viper laser model number 37-0108 by GLD Products was positioned to shine through the top of the channel, illuminating the particles in the middle of the flow. A Phantom Miro model 320s high speed camera with a Canon 65 mm lens recorded the flow for nine total experiments, at three frequencies of 0.3, 1.3, 2.3 Hz, and at three positions (top, middle, and bottom of the channel). Once recording was finished, analysis was done using the Matlab tool PIVLab. We wrote a Matlab script to separate each video into individual frames and convert each frame to greyscale to speed up the PIVLab process. A region of interest was established and each frame was processed in PIVLab before analysis. Stills from video showing the bottom of the channel when sniffing at 0.3 and 2.3 Hz can be seen in Fig. 5 e, f respectively.

**Simulations**. The flow simulations are conducted using COMSOL Multiphysics in two dimensions. The chamber is represented as a rectangle with dimensions 30 cm × 2 cm. The entrance and exit regions are 1 cm × 2 cm rectangles to represent the tubing connected to the test chamber. The inlet condition is set as an oscillating normal inflow velocity with magnitude varying sinusoidally according to the input frequency of the trial. The outlet condition is zero atmospheric pressure. The walls of the chamber are set as no slip boundary conditions. Initially, the air in the chamber is at rest. The system utilizes the default normal sized physics-controlled mesh.

**Sniffing scaling models**. Here, we present four models for the relationship between sniffing frequency and body size. We begin with a model that consider's the air's inertia. Sniff volumes, also known as a sniffing tidal volume, from literature[9,13,61] follow the trend $V_{sn} = 2.15 M^{0.99}$ mL ($N = 7$) where $M$ is body mass in kg. Using this scaling, a 20-kg dog inhales 42 mL of air during each sniff cycle, the same volume as a shot glass. For comparison, a mouse inhales 0.045 mL of air each sniff, the same volume as an eye-dropper drop, and an elephant inhales 4.6 L, the same volume as 1.2 gallon jugs. The control volume $V_{sn}$ in Fig. 2b denotes the sniffing tidal volume before it is inhaled into the lungs. The lung volume is generally 25 times larger than the sniff volume, as shown by Stahl's measurements of lung volume, $V_{lung} = 53.5 M^{1.06}$ mL ($N = 333$)[24]. When an animal inhales, it uses its diaphragm to apply a pressure $P$ to an airway with a cross sectional area $\pi r_t^2$ where $r_t$ is trachea hydraulic radius, which has been found in experiments by Tenney[62] to scale as $r_t = 0.0023 M^{0.4}$ (with $r_t$ in m and $M$ in kg). The force applied to the air may be written

$$F_L = P\pi r_t^2, \tag{6}$$

where we neglect any losses due to viscosity. The maximum pressure $P_{max}$ of the lungs, generated by muscular contraction, is independent of body size, and has constant peak magnitude of 10 kPa[63]. Throughout the duration of the sniff, the pressure is assumed to vary from positive to negative 10 kPa in a sinusoidal fashion. Therefore, the positive and negative mean values of the pressure waveform are $P = \pm\frac{2P_{max}}{\pi}$[64]. This pressure is sufficiently low that we can consider air to be incompressible.

With air being incompressible, we consider a volume $V_{tot}$ of air that must shift in order to accommodate a new sniffing volume $V_{sn}$ to enter the respiratory system, denoted by the short dashed blue and long dashed green lines in Fig. 2b respectively. The mass $m$ of the volume may be written as the product of the air density $\rho$ and the total volume, $V_{tot}$. The total volume of air in the respiratory system $V_{tot}$ may be written as the sum of the vital capacity $V_c = 56.7 M^{1.03}$ mL ($N = 315$)[24] and the functional residual capacity $V_r = 24.1 M^{1.13}$ mL ($N = 261$)[24]: $V_{tot} = V_c + V_r$. We approximate this sum using a power law best fit of these two trends, which yields, $V_{tot} = 83 M^{1.06}$ mL.

During a sniff, each air molecule is shifted by a distance $L = V_{sn}/(\pi r_t^2)$ during each period $1/f$. Assuming a sinusoidal motion of the air with displacement $s(t) = L\sin(2\pi f t)$ yields an acceleration $a = s''$ of magnitude $4\pi V_{sn} f^2/(r_t^2)$. By Newton's second law, the inertial force on the air may be written $F_a = ma$ where $a$ is as above and $m = \rho V_{tot}$. Together,

$$F_a = \frac{4\pi f^2 \rho V_{sn} V_{tot}}{r_t^2}. \tag{7}$$

The inertial force on the air $F_a$ equals the applied force of the lungs $F_L$, given in

Equation (6). Solving for the frequency $f$ yields our theoretical prediction for sniffing frequency which we call $f_1$:

$$f_1 = \sqrt{\frac{Pr_t^4}{4\rho V_{sn} V_{tot}}}. \tag{8}$$

We proceed by substituting scaling power laws for $r_t$, $V_{sn}$, and $V_{tot} = V_c + V_r$ into the above equation, which yields the sniffing frequency,

$$f_1 = 17 M^{-0.25}. \tag{9}$$

Our prediction $f_1$ is almost twice as high as the experimental data which shows mammals sniff at a frequency slower than their physical limits, possibly because it is too taxing on muscles to consistently operate at their maximum rate[65].

We also give a few caveats with regards to the assumption of sinusoidal pressure. Measurements indicate that larger animals maintain isometric scaling of sniffing volumes. However, according to previous work, as Womersley number increases, the volume flow rate decreases due to viscous effects[18]. Thus, larger animals may be applying larger pressures to compensate. This correction would bring our prediction closer to the experimental trend.

Previous studies of breathing have shown that breathing is not in fact sinusoidal. For instance, at its natural breathing rate of 2 Hz, a mouse will inhale and exhale within the first 200 ms and then remain still until the next breath. On the other hand, a mouse exploring its environment with a sniffing frequency of 10 Hz will be moving the air for almost the full duration of the sniff[66]. Since the goal is to create a simple first-order model, we do not attempt to capture these behavioral effects, and continue with assuming a sinusoidal pressure profile. In fact non-sinusoidal sniffing patterns are more difficult to study mathematically, but they give important rationale for the use of our GROMIT device. Since the motor is controlled by a computer, future workers may input different pressure profiles to find their benefits to sniffing.

We next present a model of the natural frequency of the respiratory system first proposed by David Leith[22] in 1983. Starting in the 1960s, breathing and panting were modeled by considering the chest cavity as a damped spring-mass oscillator. This model is based on experimental measurements of lung compliance $C$, resistance $R$, and inertance $I$ on humans and anesthetized animals. For regular respiration, the time scale of respiration relies on the respiratory system's resistance and compliance. On the other hand, for high speed sniffing, resistance is negligible compared to inertance. The resulting system has qualities similar to oscillating systems such as the forearm muscle[67], and electrical circuits[50]. Here, the predicted sniffing frequency $f_2$ can be expressed as

$$f_2 = \frac{1}{2\pi(CI)^{1/2}}, \tag{10}$$

where compliance $C$ may be written $C = 1.59 \times 10^{-5} M^{1.04}$ L Pa$^{-1}$ where $M$ is in kg, based on $N = 114$ mammals[24]. Inertance of the respiratory system is dominated by the inertance of the air in the trachea, as in our previous model. Based on the idea that inertial pressures should be invariant with body size, Leith[22,68] proposed inertance $I \sim M^{-1/2}$. Using this exponent, the prefactor can be estimated from Spells[50] who gathered $N = 15$ humans, dogs, and cats from previous workers across a decade of body mass. Using Fig. 7 of Spells' work, we extrapolate the data points to find $I = 7.84 M^{-1/2}$ Pa L$^{-1}$ s$^{-2}$. Combining these power laws, Leith's prediction yields

$$f_2 = 14 M^{-0.25}, \tag{11}$$

which corresponds to the blue dashed line in Fig. 2a. It is noteworthy that Leith's theoretical model has the same exponent as our first model Eq. (9) and a very similar prefactor (14 vs 17). Furthermore, each model relies on independent measurements: the first model $f_1$ relies on geometrical measurements, and Leith's model $f_2$ relies on pressure measurements. Their agreement suggests a consistent physical picture. Overall, these models suggests that sniffing frequency aligns with the respiratory system's natural frequency. Previously, panting was also proposed to correspond to natural frequency[49].

In our next model, we give an upper bound for the sniffing frequency and address the role of viscosity, which has not been considered in the previous two models. Dissipation by viscosity is expected to be important for two distinct flow regimes in the airways[69]. In the regime of slow air flows, slower than the regular breathing rate, inertial effects are reduced and viscous dissipation dominates. The regime of fast air flows is also potentially dissipative due to the generation of turbulent flow structures and their subsequent energy cascade down to dissipative lengthscales. To estimate the occurrence of turbulent flow structures in sniffing, we must take into account both the Womersley number Wo and the Reynolds number Re of the flow in the airways[23,70]. For the regime Wo $\gg 1$, pulsatile flows have their viscous effects confined to a Stokes boundary layer much thinner than the airways diameter[70]. However, the Womersley number of sniffing animals is at most of the order of the unity[18] and thus we instead apply a standard criterion of critical Reynolds number to determine the threshold to turbulence. The maximum Reynolds number associated with laminar flow in the airways is:

$$Re_{max} = \frac{2U_{max}r_t}{\nu}, \tag{12}$$

where $U_{max}$ is the maximal velocity of the displaced volume of air, the trachea

radius[62] $r_t = 0.0023M^{0.4}$, with $M$ in kg and $r_t$ in m, and $\nu = 1.48 \times 10^{-5}$ m$^2$ s$^{-1}$ is the kinematic viscosity of air. Evaluating the maximal velocity as that of the moving plug of air, $U_{max} \sim 2\pi f V_{sn}/(\pi r_t^2)$ where $V_{sn} = 2.15M^{0.99}$ mL is the sniffing volume, leads to a relationship between the sniffing frequency and maximal Reynolds number

$$f_{max} = \frac{\nu \text{Re}_{max} r_t}{4 V_{sn}}. \tag{13}$$

As shown by the experiments of Winter and Nerem in 1984, turbulence[23] in the airways will be unlikely if $\text{Re}_{max} < 2000$, which can be therefore written in terms of maximal sniffing frequency: $f_3 < f_{max}$ so that

$$f_3 < 7.91M^{-0.60}, \tag{14}$$

with $M$ in kg and $f_3$ in Hz. As shown in Fig. 2, the blue long dotted line is above all observed animal sniffing frequencies except for animals of mass larger than 30 kg, such as the horse, giraffe, and elephant. This model strengthens our confidence in neglecting viscous dissipation in our originally proposed model for smaller mammals, and simply balancing air inertia and lung force.

Lastly, we present a model based on work by Loudon and Tordesillas[18], who sought to characterize unsteady flow situations similar to those experienced during a sniff. In their model, the amplitude of an oscillating volume flow rate, $Q$, is related to the maximum pressure $P$, the radius of the channel $r_t$, the kinematic viscosity $\mu$, and the Womersley number Wo by the equation

$$Q \approx \frac{2Pr_t^3}{\mu \text{Wo}^2}. \tag{15}$$

Using a flow rate approximated as the sniff frequency times the total volume of air in the respiratory system $Q = fV_{tot}$ and Womersley number according to equation (2), equation (15) can be solved for frequency $f_4$ to be

$$f_4 = \sqrt{\frac{Pr_t}{\rho \pi V_{tot}}}. \tag{16}$$

Evaluating equation (16) with a pressure $P$ of 10 kPa[63], a trachea radius $r_t = 0.0023M^{0.4}$ m[62], air density, and respiratory volume $V_{tot} = 83M^{1.06}$ mL produces a frequency

$$f_4 = 470M^{-0.34} \tag{17}$$

in Hz as shown as a blue dot-dashed line of Fig. 2a. This trend line is more than an order of magnitude above the experimental data, indicating that Loudon's assumption of an infinite channel does not well-match the finite channel of the trachea.

## Data availability

Source data are provided with this paper. The datasets generated during and/or analyzed during the current study are available in the SniffingNatCom2020 repository, https://doi.org/10.5281/zenodo.4290759

## Code availability

The code utilized to collect data during this current study are available in the SniffingNatCom2020 repository, https://doi.org/10.5281/zenodo.4290759

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

## Acknowledgements

This material is based upon work supported by the National Science Foundation Graduate Research Fellowship and the National Science Foundation Grant Number 1510884. This work was partially funded by ACCIÓ (INNOTECRD18-1-0054); AGAUR (2018LLAV00021); the Spanish MINECO program (DPI2017-89827-R); the European Research Council (H2020-780262-SHARE4RARE); and Networking Biomedical Research Centre in the subject area of Bioengineering, Biomaterials, and Nanomedicine (CIBER-BBN), initiatives of Instituto de Investigación Carlos III (ISCIII). This work received support from the Departament d'Universitats, Recerca i Societat de la Informació de la Generalitat de Catalunya (expedient 2017 SGR 952). J.F. acknowledges the support from the Serra Húnter program.

## Author contributions

T.S. and D.H. wrote the manuscript. T.S. designed and ran experiments, data analysis, simulations, visualizations, and most mathematical calculations. T.S., A.C., and J.F. designed and built GROMIT for sensor response experiments. T.S. and A.C. performed sensor response experiments. T.S., D.H., and E.V. performed mathematical calculations for biological sniffing theoretical models. D.H. supervised the study and all authors contributed to the manuscript.

## Competing interests

The authors declare no competing interests.
