## [Peer Review File · Nature Communications]

Reviewers' comments:

Reviewer #1 (Remarks to the Author):

I commend the authors on a very interesting study that seeks to better understand how sniffing improves odor detection in both nature and man-made devices. Indeed, though most animals sniff in a periodic and rhythmic fashion when sampling an odor, to date it remains unclear from a theoretical perspective why sniffing is beneficial over a continuous inhale. In this study, the authors combine fluid dynamics theory and experiments to make two notable discoveries: (i) in nature sniff frequency scales allometrically with body mass, and (ii) there is potentially an optimal sniff frequency for sampling odor. Overall, I think that the work has significant merit and is potentially quite impactful in understanding olfaction in nature and improving the design of artificial chemical sensing devices. However, I do have several questions that concern the theoretical and experimental models and their applicability to understanding chemical sensing in the mammalian nose. Below are my comments that I intend as constructive feedback. In no way do I mean to be critical of the work, as I think it has the potential to be quite impactful. Rather, I hope that my feedback might help the authors further strengthen the overall conclusions of the study.

Theoretical Model of Allometric Scaling of Sniff Frequency:

You developed a theoretical model to understand the nature of the scaling of sniff frequency with body mass. The model predicts an absolute sniff frequency that is double that of the experimental frequency, though the scaling with body mass is in the same range. You conclude that, because of this, the "force applied by the lungs is likely half the maximum during sniffing" (pg. 4). However, in reviewing the details of this theoretical model, I have several comments/questions regarding the formulation and its assumptions and how they potentially impact the results:

1. Sniff Volume as a Function of Frequency: As I understand it, your model formulation assumes that the volume inspired during a sniff (V_{sn}) is independent of the sniff frequency. This is certainly true for flow at low Womersley numbers (Wo). However, at higher frequencies when $Wo > 1$, for the same pulsatile pressure difference the induced flow rate (and thus volume of air inspired) is appreciably reduced. This is dramatically illustrated in Figure 5 of Loudon and Tordesillas (1998) and is due to the inherent inertia of the fluid and its increasing resistance to motion at high frequency. The dependence of flow rate on the frequency/Womersley number is illustrated in Figure 6 of Loudon and Tordesillas (1998). Your model does not apparently include this frequency dependence on the inspired volume (V_{sn}) and I wonder what influence this has on the theoretical predictions?
2. Viscous Effects: Your theoretical model is founded on the assumption that viscous effects are negligible. However, in the internal conduits of the nasal passageways in mammals, I believe that there are significant losses due to viscous effects. Further, these effects are not constant and are a function of the sniff frequency, or Womersley number, as illustrated in Figure 7 of Loudon and Tordesillas (1998). Because of this, I wonder what influence the neglect of viscous effects has on your theoretical model predictions?
3. Energetic Cost of Sniffing: Your proposed theory does not incorporate the mechanics of the respiratory system, as it implicitly assumes that the diaphragm can be driven at any frequency irrespective of the energetic cost to the animal. Your theory predicts maximum sniff frequencies that are about double those measured in experiments. I wonder if part of this discrepancy is due to the neglect of the energetic cost of sniffing to the animal. Previous work by Craven et al. (2010) found that dogs sniff at a frequency of about 5 Hz, which is about the same frequency at which dogs pant based on the measurements of Crawford (1962). Interestingly, Crawford (1962) also measured the natural resonant frequency of the respiratory system in the same dogs and found it to be about 5 Hz and he discusses the energetic benefits of panting at this frequency. In Craven et al. (2010), we proposed that dogs might sniff at the same frequency as panting for the same reason—i.e., the natural frequency of the diaphragm is tuned to operate most efficiently at this frequency. I wonder if the neglect of energetic costs in your theoretical formulation might be the reason that your results overpredict the experiments? Would it be possible to potentially account for such energetic costs as a constraint in your theoretical model? If not, perhaps this may be an explanation for why the theoretical model overpredicts the experiments?

Theoretical Model of Vapor Transport and Deposition:

You have developed a mathematical model of vapor transport and deposition in a circular tube to better understand the mass transport phenomena associated with chemical sensing in your GROMIT device. This model is based on the classical Womersley solution for pulsatile flow in a tube combined with a simplified model for cross-stream molecular diffusion of vapor. You use the mathematical model to derive a variable, defined as the product of Wo and the vapor concentration ($Wo C$), with which you plot your experimental results. I agree that this type of analysis is invaluable in properly understanding the physics of the problem. However, I have a few comments/questions regarding the model formulation and assumptions and how they potentially affect the results.

1. Quasi-Steady Diffusion: As I understand it, your mathematical model for diffusion of vapor assumes a quasi-steady diffusive process at each time with no time-dependence on the concentration field at prior times. But, I believe that for the pulsatile flows you are considering, the diffusion time scale is potentially much longer than the time scale for the pulsatile flow. If so, what influence does this assumption have on the results of the mathematical model? Additionally, this treatment of diffusion does not account for the interaction between the velocity and concentration fields that causes Taylor-Aris dispersion, which is effectively an enhanced diffusivity. Do the authors think that the neglect of species dispersion has any effect on the results of the mathematical model? Though, it would certainly be more complicated, I wonder if it is possible to derive a more exact solution for the deposition following some historical mathematical studies in this area (e.g., Aris 1960; Smith 1982; Ng 2005)?

2. Dependence on Concentration: The new variable you define based on your mathematical model includes the concentration in dimensional form. Do the authors think that it is possible to collapse their results by normalizing the concentration to define a relevant non-dimensional quantity? Perhaps it is not possible in this case because the theory implicitly includes a non-linear relationship for the response of the sensor as a function of concentration. If so, it would be helpful to explain this fact. Otherwise, since the mass transport governing equation is linear in concentration and if your sensor response is linear with concentration, I would think it is possible to normalize the data by the vapor concentration.

3. Other Non-Dimensional Parameters: In this study, you only consider a single odor and the experiments are performed at a set volumetric flow rate per actuation. I believe that is why there is no need to account for different odors and variable flow rates in your theory. However, as animals in nature sniff different odors at variable flow rates, for generalizability have the authors considered including non-dimensional variables that include odorant diffusivity and flow rate—e.g., in the form of a Schmidt number and Reynolds number, respectively? Indeed, the main outcome of your mathematical model (Eq. 14 in Supplementary Information) effectively includes the diffusivity through the term Nd . Though, since you only consider a single vapor, there was no need for you to collapse the data by forming a relevant non-dimensional parameter. However, in your more generalized theory for “optimal sniffing” (on pg. 8) you do not account for the possibility of sniffing different odors at variable flow rates. Your analysis in this section implies that optimal sniffing is only a function of Wo . But, animals generally also can modulate the flow rate (e.g., see Youngentob et al. 1987; Craven et al. 2010). Thus, to generalize your “optimal sniffing” theory, I wonder if you should incorporate the possibility of sniffing different odors at variable flow rates? In the very least, I think the authors should discuss these other parameters that affect odor sampling.

Influence of Mucus Layer:

I think that perhaps one limitation of the study with regards to its generalizability to the biological system is the neglect of the mucus layer on the nasal airways that plays a critical role in the transport of odorants to the olfactory sensory neurons. To illustrate, different odors have different solubilities in mucus and, because of this, as they flow over the mucus-lined airways different odors are deposited in different locations, a phenomenon known as odorant partitioning or differential odorant sorption. Also, it takes some finite time

for odorant molecules to diffuse through the thickness of the mucus layer. Both of these aspects appreciably influence olfactory odorant transport. I certainly appreciate the difficulty of including a synthetic mucus layer in a laboratory experiment, especially given all of the unknowns about the biological mucus layer (e.g., heterogeneity, chemical composition). Incorporating a mucus layer in the theory is also very challenging. Nonetheless, I think it would be helpful if the authors make clear that the system they are studying is somewhat simplified compared to the biological system in that it does not incorporate a mucus layer.

Other Physical Constraints:

Throughout the manuscript you mention that the primary physical constraint on sniff frequency is imposed by the lungs. However, I can think of several other physical constraints on the sniff frequency in animals that are not mentioned including:

1. **Upstream Non-Sensory Length in the Nasal Cavity:** The olfactory region in the mammals is not located at the front of the nose. In keen-scented animals in particular, the sensory region is located in a posterior cul-de-sac known as the “olfactory recess” that is in the rear of the nasal cavity. Thus, it takes some finite time for inspired odorant to reach the olfactory region. In most mammals, sniffing consists of a sequential inspiration and expiration. Thus, I believe that the time scale for the transport of odorant from the nostril to the olfactory region fundamentally constrains the sniff frequency. That is, the sniff frequency has to be low enough to allow enough time for odorants to be transported to the olfactory region.

2. **Diffusion Time Scale Through the Mucus Layer:** Additionally, I believe that the sniff frequency is also somewhat constrained by the diffusion time scale through the mucus layer because it takes some finite time for odorant molecules that are deposited on the surface of the mucus layer to diffuse through it to reach the underlying olfactory receptors located on the cilia of olfactory sensory neurons. Given a typical thickness of the mucus layer in the range of 5–10 μm and estimated diffusion coefficients for some common odorants in mucus (e.g., see Table 1 in Rygg et al. 2017), the diffusion time scale is likely $\sim 0.05 - 0.1$ seconds. Although there are some complexities associated with how the concentration varies in the air phase due to the unidirectional nature of airflow in the olfactory region of many animals, I think this diffusion time scale also partially constrains the upper limit of sniff frequency in animals.

3. **Natural Frequency of the Respiratory System:** Lastly, as mentioned above, the mechanics of the respiratory system may fundamentally limit how quickly an animal can sniff. That is, sniffing at appreciably higher frequencies than the natural frequency of the respiratory system is energetically inefficient and prohibitive.

Idealization of the Womersley Solution for the Device:

You mentioned that the Womersley solution is for a circular tube, whereas your device has a square cross-section. One other—likely minor—difference is that the Womersley solution is for “fully developed” flow where there is no streamwise variation of velocity. The channel for your device is not incredibly long, so I suspect that that the flow is not truly fully developed. You could also mention this in the Discussion when describing the limitations of using the Womersley solution for your device.

References:

Aris R (1960) On the dispersion of a solute in pulsating flow through a tube. *Proceedings of the Royal Society of London A: Mathematical, Physical and Engineering Sciences* 259:370–376.

Craven B, Paterson E, Settles G (2010) The fluid dynamics of canine olfaction: Unique nasal airflow patterns as an explanation of macrosmia. *Journal of The Royal Society Interface* 7:933–943.

Crawford J EC (1962) Mechanical aspects of panting in dogs. *Journal of Applied Physiology* 17:249–251.

Loudon C, Tordesillas A (1998) The use of the dimensionless Womersley number to characterize the unsteady nature of internal flow. *Journal of Theoretical Biology* 191:63–78.

Ng CO (2005) Dispersion in steady and oscillatory flows through a tube with reversible and irreversible wall reactions. *Proceedings of the Royal Society A: Mathematical, Physical and Engineering Sciences* 462:481–515.

Rygg AD, Van Valkenburgh B, Craven BA (2017) The influence of sniffing on airflow and odorant deposition in the canine nasal cavity. *Chemical Senses* 42:683–698.

Smith R (1982) Contaminant dispersion in oscillatory flows. *Journal of Fluid Mechanics* 114:379–398.

Youngentob SL, Mozell MM, Sheehe PR, Hornung DE (1987) A quantitative analysis of sniffing strategies in rats performing odor detection tasks. *Physiology & Behavior* 41:59–69.

Please feel free to contact me if you have any specific questions about my comments.

-Brent Craven (brent.craven@fda.hhs.gov)

Reviewer #2 (Remarks to the Author):

This manuscript examines how biomimetic patterns of chemical sampling, i.e., sniffing, might be used to improve the performance of chemical sensors. In the study, the authors explore 'sniffing' using a device to generate sinusoidal airflow patterns over a chemical detector, and also use fluid dynamic simulations to examine how different sniffing frequencies could impact odor signal detection across different species. The study provides some interesting insights, mainly by illustrating the tradeoff between enhancing airflow versus reducing signal-to-noise ratio by sniffing at different rates. However, it is not clear that these insights could not be arrived at via the fluid mechanic calculations alone (which are standard), nor to what degree the calculations or the data can be generalized to different biological systems. Thus overall I feel that this study has a fairly limited impact - though it would be worthwhile to publish in a more specialized journal. Major comments are below.

1. The authors make several simplifications in their approach that seriously limit the applicability of their findings to different biological systems. One is that they treat the nasal cavity as a uniform space with cylindrical or square geometry, which of course it is not. This assumption is mentioned in the Discussion, but this does not resolve its limitation. The highly convoluted internal structure of the mammalian nasal cavity likely leads to drastically different airflow patterns - i.e., Womersley numbers - than estimated here, in ways that are not at all intuitive. Thus one cannot easily extrapolate from their results to those of an actual mammal.
2. The authors also presume that sniffing is sinusoidal across frequencies, which is not the case. At the lower end of sniffing frequencies, sniff duration does not increase as sniff interval increases, and likewise the rate of change in intranasal pressure (and thus airflow velocity) does not change linearly with frequency (see, for example, Fig. 6A in Diaz-Quesada et al., *J Neurosci* 2018). This impacts the basic calculations done in the paper with respect to number of molecules inhaled per unit time, molecules sampled per sniff, etc.
3. The authors conclude that, at higher sniff frequencies, rapid sniffing is disadvantageous because the number of inhaled/detected molecules falls below the noise level of the sensor. However, an obvious solution to this problem would be to simply average 'responses' to multiple sniffs, which would increase the

signal:noise ratio as the square root of the number of sniffs. Surprisingly, this possibility was not raised in the Discussion.

Reviewer #3 (Remarks to the Author):

What are the major claims of the paper? The authors examine sniffing parameters for small, medium, and large animals and suggest that fast sniffing enables animals to recognize an odor quickly. Slow sniffing takes less energy, increases signal amplitude, but delays the odor recognition process. They suggest that their findings may help electronic olfaction sensor designers modulate sniffing frequency according to the odors being measured.

Are they novel and will they be of interest to others in the community and the wider field? They give some odor flow simulation results that show laminar flow near the edges of their rectangular flow channel. This finding is well known in the community. They do not include any nasal turbinate contours but acknowledge that they should be included in future work. They give a very nice historical review of past studies. They merge published results with experiments of their own. They validate that historical mathematical models are still valid and can be used on modern miniaturized electronic sniffing devices.

Do you feel that the paper will influence thinking in the field? This paper could be a reference for International Standards that give guidelines for designing machine olfaction devices. New sniffing algorithms tailored to specific applications could follow from their work.

The title is a bit misleading. I do not see any mechanism to redirect the odor for onto the sensors. I think something like this would be more appropriate:

"Sniffing speeds up chemical detection by controlling air-flows near sensors"

Reviewer #1 (Remarks to the Author):

I commend the authors on a very interesting study that seeks to better understand how sniffing improves odor detection in both nature and man-made devices. Indeed, though most animals sniff in a periodic and rhythmic fashion when sampling an odor, to date it remains unclear from a theoretical perspective why sniffing is beneficial over a continuous inhale.

Thank you for your enthusiasm for this manuscript.

In this study, the authors combine fluid dynamics theory and experiments to make two notable discoveries: (i) in nature sniff frequency scales allometrically with body mass, and (ii) there is potentially an optimal sniff frequency for sampling odor.

This is an excellent summary.

Overall, I think that the work has significant merit and is potentially quite impactful in understanding olfaction in nature and improving the design of artificial chemical sensing devices. However, I do have several questions that concern the theoretical and experimental models and their applicability to understanding chemical sensing in the mammalian nose. Below are my comments that I intend as constructive feedback. In no way do I mean to be critical of the work, as I think it has the potential to be quite impactful. Rather, I hope that my feedback might help the authors further strengthen the overall conclusions of the study.

We appreciate your insightful comments, which have improved the manuscript on many fronts.

Theoretical Model of Allometric Scaling of Sniff Frequency:

You developed a theoretical model to understand the nature of the scaling of sniff frequency with body mass. The model predicts an absolute sniff frequency that is double that of the experimental frequency, though the scaling with body mass is in the same range. You conclude that, because of this, the "force applied by the lungs is likely half the maximum during sniffing" (pg. 4). However, in reviewing the details of this theoretical model, I have several comments/questions regarding the formulation and its assumptions and how they potentially impact the results:

1. Sniff Volume as a Function of Frequency: As I understand it, your model formulation assumes that the volume inspired during a sniff (V_{sn}) is independent of the sniff frequency.

Indeed, sniff volume scales linearly with body mass.

This is certainly true for flow at low Womersley numbers (Wo). However, at higher frequencies when $Wo > 1$, for the same pulsatile pressure difference the induced flow rate (and thus volume of air inspired) is appreciably reduced. This is dramatically illustrated in Figure 5 of Loudon and Tordesillas (1998) and is due to the inherent inertia of the fluid and its increasing resistance to motion at high frequency.

The dependence of flow rate on the frequency/Womersley number is illustrated in Figure 6 of Loudon and Tordesillas (1998). Your model does not apparently include this frequency dependence on the inspired volume (V_{sn}) and I wonder what influence this has on the theoretical predictions?

Indeed, at higher frequencies above $Wo = 1$, the same applied pulsatile pressure difference induces a reduced flow rate as Loudon and Tordesillas (1998) showed. Loudon showed that due to increasing viscous effects with Wo , the flow rate scales as Wo^{-1} .

Nevertheless, animals appear to be keeping sniffing volume linear with body mass in the regime we observed. This suggests that they are able to apply larger pressure to

compensate for the increased viscous forces. It's possible that the largest animals like the elephant may not be able to apply adequate pressures to compensate. We have applied Loudon's Volume- W_0 relationship from their 1998 paper for completeness. The prediction is far from the experiments, which suggests modeling trachea as an infinite pipe is not as accurate as our modeling efforts.

For making our device, our assumption of constant sniff volume remains valid because we maintain the amplitude of the physical plunger through direct control of the stepper motor. This control ensures that the same volume of air is inspired/expired each cycle independent of sniff frequency. To do so, the stepper motor must pull harder on the diaphragm as W_0 increases, and likely uses more power. This may limit future devices and we now mention it starting on line 485.

2. Viscous Effects: Your theoretical model is founded on the assumption that viscous effects are negligible. However, in the internal conduits of the nasal passageways in mammals, I believe that there are significant losses due to viscous effects. Further, these effects are not constant and are a function of the sniff frequency, or Womersley number, as illustrated in Figure 7 of Loudon and Tordesillas (1998). Because of this, I wonder what influence the neglect of viscous effects has on your theoretical model predictions?

We now include a new model f_3 that shows when viscous effects become important. This model is based on experiments conducted by Winter and Nerem in in the 1980s.

Dissipation by viscosity is expected to be important for two very different flow regimes in the airways. First, in the regime of very slow air flows, below the regular breathing rate, inertial effects are reduced and viscous dissipation naturally occurs. Second, the regime of very fast air flows is also highly dissipative, because of the generation of turbulent flow structures and the subsequent turbulent cascade down to dissipative length scales.

Our model gives the sniffing frequency scaling so that the sniff remains below a limiting turbulent Reynolds number, where viscosity would severely slow the sniff.

3. Energetic Cost of Sniffing: Your proposed theory does not incorporate the mechanics of the respiratory system, as it implicitly assumes that the diaphragm can be driven at any frequency irrespective of the energetic cost to the animal. Your theory predicts maximum sniff frequencies that are about double those measured in experiments. I wonder if part of this discrepancy is due to the neglect of the energetic cost of sniffing to the animal. Previous work by Craven et al. (2010) found that dogs sniff at a frequency of about 5 Hz, which is about the same frequency at which dogs pant based on the measurements of Crawford (1962). Interestingly, Crawford (1962) also measured the natural resonant frequency of the respiratory system in the same dogs and found it to be about 5 Hz and he discusses the energetic benefits of panting at this frequency. In Craven et al. (2010), we proposed that dogs might sniff at the same frequency as panting for the same reason—i.e., the natural frequency of the diaphragm is tuned to operate most efficiently at this frequency.

Thank you for these citations which we now include in the manuscript.

I wonder if the neglect of energetic costs in your theoretical formulation might be the reason that your results overpredict the experiments? Would it be possible to potentially account for such energetic costs as a constraint in your theoretical model? If not, perhaps this may be an explanation for why the theoretical model overpredicts the experiments?

Thanks for bringing up this classic literature on panting and resonant frequency of the respiratory system. We now include a new model f_2 , first proposed by David Leith, that relies on the measured compliance and inertance of mammal respiratory systems. The new

model corroborates our previous model and links our work to a wealth of literature from the 1960s on respiratory systems.

We also discuss energetic costs of sniffing at high frequency and bring up the points about energy consideration using devices at high speed. This explanation starts on line 475.

Theoretical Model of Vapor Transport and Deposition:

You have developed a mathematical model of vapor transport and deposition in a circular tube to better understand the mass transport phenomena associated with chemical sensing in your GROMIT device. This model is based on the classical Womersley solution for pulsatile flow in a tube combined with a simplified model for cross-stream molecular diffusion of vapor. You use the mathematical model to derive a variable, defined as the product of Wo and the vapor concentration ($Wo C$), with which you plot your experimental results. I agree that this type of analysis is invaluable in properly understanding the physics of the problem.

We agree with this summary.

However, I have a few comments/questions regarding the model formulation and assumptions and how they potentially affect the results.

1. Quasi-Steady Diffusion: As I understand it, your mathematical model for diffusion of vapor assumes a quasi-steady diffusive process at each time with no time-dependence on the concentration field at prior times. But, I believe that for the pulsatile flows you are considering, the diffusion time scale is potentially much longer than the time scale for the pulsatile flow. If so, what influence does this assumption have on the results of the mathematical model?

We agree, the diffusion time scale is much longer than the pulsatile time scale. This is why only a very thin region of the molecules have the chance to land on the sensor surface. But we like your explanation so we also include it in the main text. This we explained starting on line 136 and in more detail starting on line 88 of the supplement.

Additionally, this treatment of diffusion does not account for the interaction between the velocity and concentration fields that causes Taylor-Aris dispersion, which is effectively an enhanced diffusivity. Do the authors think that the neglect of species dispersion has any affect on the results of the mathematical model? Though, it would certainly more complicated, I wonder if it is possible to derive a more exact solution for the deposition following some historical mathematical studies in this area (e.g., Aris 1960; Smith 1982; Ng 2005)?

Thank you for bringing up this point which we had not considered. Our experiment using GROMIT did not show a significant change in the period or amplitude of the sensor current with time, indicating that, at least for the parameters studied, the effects of axial diffusion are minimal. Future devices running at higher speed may find that Taylor-Aris dispersions is more dominant, which we address starting on line 176.

2. Dependence on Concentration: The new variable you define based on your mathematical model includes the concentration in dimensional form. Do the authors think that it is possible to collapse their results by normalizing the concentration to define a relevant non-dimensional quantity? Perhaps it is not possible in this case because the theory implicitly includes a non-linear relationship for the response of the sensor as a function of concentration. If so, it would be helpful to explain this fact. Otherwise, since the mass transport governing equation is linear in concentration and if your sensor response is linear with concentration, I would think it is possible to normalize the data by the vapor concentration.

We had previously tried to normalize the amplitude data before, however the highly fluctuating response of the sensors to different concentration values presents a relationship that cannot be determined with any certainty. In the pasted figure below, the Amplitude is non-dimensionalized by dividing by Beta and concentration. The solid theory line does seem to follow the trend, but the large error bars are not desirable. More importantly, the “noise threshold” lines in blue are a function of the concentration and therefore are at different locations for each set of data points (see below figure). Expressed in this collapsed form it is difficult to see which data points align to which minimum noise threshold. Additionally, the y axis in the normalized form includes a fitting factor, Beta, specific to the sensors chosen. This inclusion limits the generalizability of the results. Therefore we elected to present the data in dimensional form because it is easier for the reader to see, and it keeps the error a reasonable value.

Response Fig 1: The relationship between Wormsley number and the dimensionless signal amplitude.

3. Other Non-Dimensional Parameters: In this study, you only consider a single odor and the experiments are performed at a set volumetric flow rate per actuation. I believe that is why there is no need to account for different odors and variable flow rates in your theory. However, as animals in nature sniff different odors at variable flow rates, for generalizability have the authors considered including non-dimensional variables that include odorant diffusivity and flow rate—e.g., in the form of a Schmidt number and Reynolds number, respectively?

Thanks for the suggestion. We now define Schmidt number in the paper and show that the target material, ethanol, is sufficiently diffusive for our model to apply. This will also convey to the reader what target materials they might use in future devices. This discussion starts on line 165.

Indeed, the main outcome of your mathematical model (Eq. 14 in Supplementary Information) effectively includes the diffusivity through the term Nd . Though, since you only consider a single vapor, there was no need for you to collapse the data by forming a relevant non-dimensional parameter.

Correct. With only one kind of vapor, we only need to keep track of the number of molecules in the relevant volume that has time to diffuse onto the sensors. We imagine that this theory may also work with several vapors each with their own diffusive volume. As long as the vapors don't interact, we could consider their landing on the sensor to be independent.

However, in your more generalized theory for “optimal sniffing” (on pg. 8) you do not account for the possibility of sniffing different odors at variable flow rates.

Yes, for simplicity, we only consider a single odor source.

Your analysis in this section implies that optimal sniffing is only a function of W_o . But, animals generally also can modulate the flow rate (e.g., see Youngentob et al. 1987; Craven et al. 2010). Thus, to generalize your “optimal sniffing” theory, I wonder if you should incorporate the possibility of sniffing different odors at variable flow rates? In the very least, I think the authors should discuss these other parameters that affect odor sampling.

This is a good point. We agree that in nature, likely more than one odor will be present. We now state that that different odors will have different optimal rates, and that animals may need to sniff at different frequencies to optimally drawn in different odors. These statements start on line 434.

Influence of Mucus Layer:

I think that perhaps one limitation of the study with regards to its generalizability to the biological system is the neglect of the mucus layer on the nasal airways that plays a critical role in the transport of odorants to the olfactory sensory neurons.

To illustrate, different odors have different solubilities in mucus and, because of this, as they flow over the mucus-lined airways different odors are deposited in different locations, a phenomenon known as odorant partitioning or differential odorant sorption. Also, it takes some finite time for odorant molecules to diffuse through the thickness of the mucus layer. Both of these aspects appreciably influence olfactory odorant transport. I certainly appreciate the difficulty of including a synthetic mucus layer in a laboratory experiment, especially given all of the unknowns about the biological mucus layer (e.g., heterogeneity, chemical composition). Incorporating a mucus layer in the theory is also very challenging. Nonetheless, I think it would be helpful if the authors make clear that the system they are studying is somewhat simplified compared to the biological system in that it does not incorporate a mucus layer.

We agree that biological systems have mucus layers which play a critical role in protection of the respiratory system and transporting odorants to the olfactory sensor neurons. For simplicity a first sniffing device like ours should not be designed for full-on mimicry. A dry system is easier to model and interpret.

The reviewer brings a good point that our model and system must be carefully applied to understand more complex biological systems. We add this caveat on line 435 as well as the diffusion rates of mucus starting on line 455.

Other Physical Constraints:

Throughout the manuscript you mention that the primary physical constraint on sniff frequency is imposed by the lungs. However, I can think of several other physical constraints on the sniff frequency in animals that are not mentioned including:

1. Upstream Non-Sensory Length in the Nasal Cavity: The olfactory region in the mammals is not located at the front of the nose. In keen-scented animals in particular, the sensory region is located in a posterior cul-de-sac known as the “olfactory recess” that is in the rear of the nasal cavity. Thus, it takes some finite time for inspired odorant to reach the olfactory region. In most mammals, sniffing consists of a sequential inspiration and expiration. Thus, I believe that the time scale for the transport

of odorant from the nostril to the olfactory region fundamentally constrains the sniff frequency. That is, the sniff frequency has to be low enough to allow enough time for odorants to be transported to the olfactory region.

We agree that there is a potential upper limit on sniff frequency due to the fact that odorants must be able to reach the rear of the nasal cavity each sniff to be recognized. For dogs as an example, with an average velocity through the nasal cavity on the order of 5 m/s, and a snout length on the order of 0.1 m, the period of a sniff must be at least $0.1/20 = 0.005$ seconds. This ensures the frequency must be less than 200 Hz which is an order of magnitude above the observed frequency. We explain this starting on line 455.

2. Diffusion Time Scale Through the Mucus Layer: Additionally, I believe that the sniff frequency is also somewhat constrained by the diffusion time scale through the mucus layer because it takes some finite time for odorant molecules that are deposited on the surface of the mucus layer to diffuse through it to reach the underlying olfactory receptors located on the cilia of olfactory sensory neurons. Given a typical thickness of the mucus layer in the range of 5–10 μm and estimated diffusion coefficients for some common odorants in mucus (e.g., see Table 1 in Rygg et al. 2017), the diffusion time scale is likely $\sim 0.05 - 0.1$ seconds. Although there are some complexities associated with how the concentration varies in the air phase due to the unidirectional nature of airflow in the olfactory region of many animals, I think this diffusion time scale also partially constrains the upper limit of sniff frequency in animals.

We agree that the diffusion through the mucus layer is important for the sensing capabilities of mammals. However, the time for molecules to diffuse through the mucus layer should be independent of the airflow parameters. This would cause a phase lag in the input to the olfactory receptors but should not affect the molecules path onto the top of the mucus layer. We elaborate on the mucus diffusion in the mucus discussion starting on line 460.

3. Natural Frequency of the Respiratory System: Lastly, as mentioned above, the mechanics of the respiratory system may fundamentally limit how quickly an animal can sniff. That is, sniffing at appreciably higher frequencies than the natural frequency of the respiratory system is energetically inefficient and prohibitive.

We agree that another potential limit in the same range as the experimental sniff frequency is due to energetic constraints due to the natural frequency of the respiratory system. If the sniff frequency deviates too high above this (5 Hz for dogs) natural frequency then it could be too energetically costly to maintain. This may also be a hypothesis for why quicker bouts of sniffing are only observed for a limited time. We did not include these energetic constraints in our model since the electronic nose mimic is made with a diaphragm of plastic and latex exhibiting a different resonant frequency to that of a natural respiratory system. We add discussion of this concern starting on line 475.

Additionally, we present a new scaling argument that matches well with the experimental findings using the relation between compliance, inertance, and frequency. This scaling calculation starts on line 367.

Idealization of the Womersley Solution for the Device:

You mentioned that the Womersley solution is for a circular tube, whereas your device has a square cross-section. One other—likely minor—difference is that the Womersley solution is for “fully developed” flow where there is no streamwise variation of velocity. The channel for your device is not incredibly long, so I suspect that that the flow is not truly fully developed. You could also mention this in the Discussion when describing the limitations of using the Womersley solution for your device.

We thank the reviewer for this suggestion and have incorporated the line, "Also, our theory assumes fully developed flow yet our experiments do not utilize a channel that is over 10x the diameter to ensure a fully developed profile" starting on line 492.

References:

Aris R (1960) On the dispersion of a solute in pulsating flow through a tube. Proceedings of the Royal Society of London A: Mathematical, Physical and Engineering Sciences 259:370–376.

Craven B, Paterson E, Settles G (2010) The fluid dynamics of canine olfaction: Unique nasal airflow patterns as an explanation of macrosmia. Journal of The Royal Society Interface 7:933–943.

Crawford J EC (1962) Mechanical aspects of panting in dogs. Journal of Applied Physiology 17:249–251.

Loudon C, Tordesillas A (1998) The use of the dimensionless Womersley number to characterize the unsteady nature of internal flow. Journal of Theoretical Biology 191:63–78.

Ng CO (2005) Dispersion in steady and oscillatory flows through a tube with reversible and irreversible wall reactions. Proceedings of the Royal Society A: Mathematical, Physical and Engineering Sciences 462:481–515.

Rygg AD, Van Valkenburgh B, Craven BA (2017) The influence of sniffing on airflow and odorant deposition in the canine nasal cavity. Chemical Senses 42:683–698.

Smith R (1982) Contaminant dispersion in oscillatory flows. Journal of Fluid Mechanics 114:379–398.

Youngentob SL, Mozell MM, Sheehe PR, Hornung DE (1987) A quantitative analysis of sniffing strategies in rats performing odor detection tasks. Physiology & Behavior 41:59–69.

Thanks for these references, which we now cite in the manuscript.

Please feel free to contact me if you have any specific questions about my comments.

-Brent Craven (brent.craven@fda.hhs.gov)

Thanks Dr. Craven for your heroic review which substantially improved the completeness of our manuscript. We appreciate your support and we look forward to any future revisions you might have.

Reviewer #2 (Remarks to the Author):

This manuscript examines how biomimetic patterns of chemical sampling, i.e., sniffing, might be used to improve the performance of chemical sensors. In the study, the authors explore 'sniffing' using a device to generate sinusoidal airflow patterns over a chemical detector, and also use fluid dynamic simulations to examine how different sniffing frequencies could impact odor signal detection across different species.

The study provides some interesting insights, mainly by illustrating the tradeoff between enhancing airflow versus reducing signal-to-noise ratio by sniffing at different rates. However, it is not clear that these insights could not be arrived at via the fluid mechanic calculations alone (which are standard), nor to what degree the calculations or the data can be generalized to different biological systems.

Thank you for your interest in our work. We now more properly motivate our device in the introduction. A full mathematical model of the sniffing system must take into account the diffusion of chemicals onto a sensor, fluid mechanics of the air, and the channels and walls of the system. This becomes even more complex with multiple chemical species. As a result, these three aspects have never been before tackled in a single model. Therefore, building a device like ours can help test both biology hypotheses and give validation to theoretical models.

Thus overall I feel that this study has a fairly limited impact - though it would be worthwhile to publish in a more specialized journal. Major comments are below.

The main contribution of our work is the presentation of a new device that can be used to test biological hypotheses and validate theoretical models. We believe that these results will inspire others to take this approach and provide much needed data to this field. Since there are so many workers across fields (neuroscience, computational fluid dynamics, biomechanics), publishing in Nat Com seems like the best way to reach them all.

1. The authors make several simplifications in their approach that seriously limit the applicability of their findings to different biological systems. One is that they treat the nasal cavity as a uniform space with cylindrical or square geometry, which of course it is not. This assumption is mentioned in the Discussion, but this does not resolve its limitation. The highly convoluted internal structure of the mammalian nasal cavity likely leads to drastically different airflow patterns - i.e., Womersley numbers - than estimated here, in ways that are not at all intuitive. Thus one cannot easily extrapolate from their results to those of an actual mammal.

The reviewer is correct that animal nasal cavities are complex. We take their shapes into account using the hydraulic diameter, as was done by Craven in 2010. This approach allows us to consider to some extent the complex shapes in calculation of the Womersley numbers and their flows.

In our simple back of the envelope calculation we endeavored to determine the dominant reasons why there is frequency scaling across body size. We have done so without consideration of the complex animals nasal passages. These shapes have not been fully documented across animals, which makes it difficult to incorporate them into our modeling of sniffing frequency. Nevertheless, our coarse-grained approach has captured the essence of sniffing physics as shown by the similarity in the exponents of the experimental and theoretical scaling laws. We hope that future workers will take our approach further.

We have good reasons to use a simplified nasal cavity for our device. Devices are not meant to be just mimetic of animals. They are also meant to be practical to build. Studying a simple device is clearly the best approach to start with. Future workers can 3d print these complex shapes and use our approach to test hypotheses to rationalize the complex shapes of animal. We have added an explanation starting on line 110 to show how we simplified the geometry using previously accepted methods.

2. The authors also presume that sniffing is sinusoidal across frequencies, which is not the case. At the lower end of sniffing frequencies, sniff duration does not increase as sniff interval increases, and likewise the rate of change in intranasal pressure (and thus airflow velocity) does not change linearly with frequency (see, for example, Fig. 6A in Diaz-Quesada et al., J Neurosci 2018). This impacts the basic calculations done in the paper with respect to number of molecules inhaled per unit time, molecules sampled per sniff, etc.

This is nice point and we bring up his reference in the paper. We added this caveat starting on line 353. We note that non-sinusoidal profiles are difficult to study theoretically. However, this provides additional rationale for using our device to test their benefits.

3. The authors conclude that, at higher sniff frequencies, rapid sniffing is disadvantageous because the number of inhaled/detected molecules falls below the noise level of the sensor. However, an obvious solution to this problem would be to simply average 'responses' to multiple sniffs, which would increase the signal:noise ratio as the square root of the number of sniffs. Surprisingly, this possibility was not raised in the Discussion.

We agree that in general, a solution to increase the signal to noise ratio of the sensors would be to simply average the response over multiple sniffs. This could theoretically lower the lower cut-off indicated by the lower shaded region of Figure 3d. However, it is important to remark that for such a solution, the noise needs to be random and centered in the signal mean (zero-mean noise). We only measure positive current, so the noise is not the same below the signal than above the signal when the signal amplitude is similar (and lower) than the noise level. Therefore, in this case, averaging the signal will not work. Additionally, for dynamically changing environments such as are experienced by animals, repeated measurements are often not possible. We have explained this starting on line 500.

Reviewer #3 (Remarks to the Author):

What are the major claims of the paper? The authors examine sniffing parameters for small, medium, and large animals and suggest that fast sniffing enables animals to recognize an odor quickly. Slow sniffing takes less energy, increases signal amplitude, but delays the odor recognition process. They suggest that their findings may help electronic olfaction sensor designers modulate sniffing frequency according to the odors being measured.

We agree with the reviewer's assessment of our major claims. Thank you for your comments.

Are they novel and will they be of interest to others in the community and the wider field? They give some odor flow simulation results that show laminar flow near the edges of their rectangular flow channel. This finding is well known in the community. They do not include any nasal turbinate contours but acknowledge that they should be included in future work. They give a very nice historical review of past studies. They merge published results with experiments of their own. They validate that historical mathematical models are still valid and can be used on modern miniaturized electronic sniffing devices.

We thank the reviewer for the positive review of our work.

Do you feel that the paper will influence thinking in the field? This paper could be a reference for International Standards that give guidelines for designing machine olfaction devices. New sniffing algorithms tailored to specific applications could follow from their work.

We thank the reviewer for their praise of the work and agree that this should serve as a guideline for designing machine olfaction devices.

The title is a bit misleading. I do not see any mechanism to redirect the odor for onto the sensors. I think something like this would be more appropriate:

"Sniffing speeds up chemical detection by controlling air-flows near sensors"

We thank the reviewer for suggesting the new title, which we are delighted to use in the paper.

REVIEWER COMMENTS

Reviewer #1 (Remarks to the Author):

I thank the authors for thoroughly responding to my previous comments. Overall, they have addressed my concerns. I do have several suggestions below to consider. But, otherwise, I believe the manuscript is acceptable. I commend the authors on a very nice study that I believe will be very impactful to the fields of olfaction and artificial chemical sensing.

Comments:

1. Diffusion Time Scale Through the Mucus Layer: Beginning on line 455 of the revised manuscript you describe an order of magnitude analysis for the diffusion time scale of a molecule through mucus that has implications for interpreting sniff frequency in animals as you describe. But, I think there is a slight mistake in the analysis. I agree with your supplemental equation 3 for estimating the diffusion time scale. However, I believe that the diffusion coefficient for many odorous vapors is larger than $1 \times 10^{-10} \text{ m}^2/\text{s}$. Most odors that we have considered have a diffusion coefficient of at least $5 \times 10^{-10} \text{ m}^2/\text{s}$ and it is often closer to $1 \times 10^{-9} \text{ m}^2/\text{s}$ (e.g., see Table 1 in Rygg et al. 2017 and Table 1 in Lawson et al. 2012). Using coefficients in this range gives diffusion time scales on the order of 0.1–0.05 seconds, whereas you are reporting a diffusion time scale of roughly 0.5 seconds. This difference is potentially very important with respect to interpreting the influence of molecular diffusion through mucus on olfaction. As you suggest, if the diffusion time scale is truly 0.5 seconds this means that dogs and other animals that sniff quickly (in the 5-10 Hz range) would require several sniffs before they could detect an odor. However, there is research showing that animals can detect odors within a single sniff (e.g., see Uchida and Mainen, *Nature Neuroscience*, 2003, 6:1224–1229). This implies that the diffusion time scale must be much faster than this. Using a better estimate of the diffusion coefficient, I think the diffusion time scale is likely in the 0.1–0.05 second range. But, regardless, I agree with you that this does physically constrain the frequency of sniffing in animals. Note that I think there are several explanations for the “pause” that you refer to in this paragraph. One explanation is “breath stacking” which we observed in our measurements of dogs sniffing in Craven et al. 2010. That is, dogs seem to inspire a slightly larger volume of air than they expire during each sniff. In the respiratory literature this is known as “breath stacking.” Because of this, after a series of sniffs, the dog has a large volume of air accumulated in their lungs that they will then expire. Though it is not widely documented in the literature, it is widely observed by many detector dog trainers and handlers that I have spoken to, and I have observed it as a pet dog owner. Anyway, this latter point regarding the pause is not all that important. But, I think that the diffusion time scale and its implications are important. If you agree with the above scaling analysis/information, I would recommend that you revise this section of the manuscript accordingly.

2. Transit Time to Olfactory Region: Beginning on line 447 of the revised manuscript you describe the physical constraint on sniff frequency due to the transit time required for odors to flow through the non-sensory portion of the nasal cavity and reach the olfactory region. I agree with your analysis and conclusions, but I think there might be a slight mistake in the calculation. Given a length, L , of 0.1 m and a representative flow speed, U , of 5 m/s, I calculate a time scale of $t = L/U = 0.02$ seconds and not the 0.0005 seconds you have indicated in this paragraph. If you could, please double check this calculation. Also, please note that this time scale is only for inspiration whereas a full sniff for most macrosmatic mammals consists of an inspiration followed by an expiration. So, the corresponding sniff frequency will include a factor of 2 to account for expiration. These are minor details that do not affect the overall conclusion, but they do affect the numerical values reported in this paragraph.

3. Minor Comment on Odor Sorption: On lines 439-440 you state that “Our study assumes a uniform distribution of collection, which may be valid for certain chemicals.” I agree. Though, I think your analysis that assumes a uniform distribution of collection is most pertinent to relatively insoluble odors because these odors are deposited rather uniformly along mucus-lined olfactory airways (e.g., see Figure 8 in Rygg et al. 2017). You could possibly revise this sentence to make this more clear.

Please feel free to contact me if you have any specific questions about my comments. Again, congratulations

on a very nice study!

-Brent Craven (brent.craven@fda.hhs.gov)

Reviewer #2 (Remarks to the Author):

The authors have expanded on their modeling efforts to address the previous comments from reviewers, and added text to address some comments from this reviewer that have to do with caveats for generalizing from their findings to a true biological situation. While it is still not clear to me whether the chief contribution of this paper is from the allometry, the modeling, or the new 'artificial sniffing' device, I will certainly defer to the other reviewers in their assessment of the modeling work. Thus, I am satisfied with these revisions.

Reviewer #3 (Remarks to the Author):

The authors have addressed my concerns.

Reviewer #1 (Remarks to the Author):

1. Diffusion Time Scale Through the Mucus Layer: Beginning on line 455 of the revised manuscript you describe an order of magnitude analysis for the diffusion time scale of a molecule through mucus that has implications for interpreting sniff frequency in animals as you describe. But, I think there is a slight mistake in the analysis. I agree with your supplemental equation 3 for estimating the diffusion time scale. However, I believe that the diffusion coefficient for many odorous vapors is larger than 1×10^{-10} m²/s. Most odors that we have considered have a diffusion coefficient of at least 5×10^{-9} m²/s and it is often closer to 1×10^{-9} m²/s (e.g., see Table 1 in Rygg et al. 2017 and Table 1 in Lawson et al. 2012). Using coefficients in this range gives diffusion time scales on the order of 0.1–0.05 seconds, whereas you are reporting a diffusion time scale of roughly 0.5 seconds. This difference is potentially very important with respect to interpreting the influence of molecular diffusion through mucus on olfaction. As you suggest, if the diffusion time scale is truly 0.5 seconds this means that dogs and other animals that sniff quickly (in the 5-10 Hz range) would require several sniffs before they could detect an odor. However, there is research showing that animals can detect odors within a single sniff (e.g., see Uchida and Mainen, Nature Neuroscience, 2003, 6:1224–1229). This implies that the diffusion time scale must be much faster than this. Using a better estimate of the diffusion coefficient, I think the diffusion time scale is likely in the 0.1–0.05 second range. But, regardless, I agree with you that this does physically constrain the frequency of sniffing in animals. Note that I think there are several explanations for the “pause” that you refer to in this paragraph. One explanation is “breath stacking” which we observed in our measurements of dogs sniffing in Craven et al. 2010. That is, dogs seem to inspire a slightly larger volume of air than they expire during each sniff. In the respiratory literature this is known as “breath stacking.” Because of this, after a series of sniffs, the dog has a large volume of air accumulated in their lungs that they will then expire. Though it is not widely documented in the literature, it is widely observed by many detector dog trainers and handlers that I have spoken to, and I have observed it as a pet dog owner. Anyway, this latter point regarding the pause is not all that important. But, I think that the diffusion time scale and its implications are important. If you agree with the above scaling analysis/information, I would recommend that you revise this section of the manuscript accordingly.

Thank you for this comment. This highlights the importance of precision in an order of magnitude calculation. Using a more precise estimate of $6.5\text{--}8.2 \times 10^{-10}$ for the diffusion coefficient provides a frequency which is in fact faster than the sniffing frequency. We have updated the text accordingly (lines 462-473).

We have also removed mention of the pause which as you indicated is a minor point and is not the key focus of this manuscript. As it is not widely documented in the literature, perhaps this could be an interesting avenue for future work.

2. Transit Time to Olfactory Region: Beginning on line 447 of the revised manuscript you describe the physical constraint on sniff frequency due to the transit time required for odors to flow through the non-sensory portion of the nasal cavity and reach the olfactory region. I agree with your analysis and

conclusions, but I think there might be a slight mistake in the calculation. Given a length, L , of 0.1 m and a representative flow speed, U , of 5 m/s, I calculate a time scale of $t = L/U = 0.02$ seconds and not the 0.0005 seconds you have indicated in this paragraph. If you could, please double check this calculation. Also, please note that this time scale is only for inspiration whereas a full sniff for most macroscopic mammals consists of an inspiration followed by an expiration. So, the corresponding sniff frequency will include a factor of 2 to account for expiration. These are minor details that do not affect the overall conclusion, but they do affect the numerical values reported in this paragraph.

Thank you for catching this typo. The numbers are now correct.

3. Minor Comment on Odor Sorption: On lines 439-440 you state that “Our study assumes a uniform distribution of collection, which may be valid for certain chemicals.” I agree. Though, I think your analysis that assumes a uniform distribution of collection is most pertinent to relatively insoluble odors because these odors are deposited rather uniformly along mucus-lined olfactory airways (e.g., see Figure 8 in Rygg et al. 2017). You could possibly revise this sentence to make this more clear.

Thank you for pointing out that we need to be more clear. We have changed the comment to “Our study assumes a uniform distribution of collection, which may be valid for certain chemicals. This assumption is most pertinent to relatively insoluble odors because these odors are deposited rather uniformly along mucus-lined olfactory airways \cite{Rygg2017-ig}.”

Reviewer #2 (Remarks to the Author):

The authors have expanded on their modeling efforts to address the previous comments from reviewers, and added text to address some comments from this reviewer that have to do with caveats for generalizing from their findings to a true biological situation. While it is still not clear to me whether the chief contribution of this paper is from the allometry, the modeling, or the new ‘artificial sniffing’ device, I will certainly defer to the other reviewers in their assessment of the modeling work. Thus, I am satisfied with these revisions.

Thank you for your support.

Reviewer #3 (Remarks to the Author):

The authors have addressed my concerns.

Thank you for your support.

Reviewers' Comments:

Reviewer #1:

Remarks to the Author:

The authors have addressed my concerns. Congratulations on an excellent study!

- Brent Craven

REVIEWERS' COMMENTS

Reviewer #1 (Remarks to the Author):

The authors have addressed my concerns. Congratulations on an excellent study!

- Brent Craven

RESPONSE

Thank you for your support and assistance through this process.